# Impact of school closures and reopening on COVID-19 caseload in 6 cities of Pakistan: An Interrupted Time Series Analysis

**Abdul Mueed**[1], **Taimoor Ahmad**[1], **Mujahid Abdullah**[1], **Faisal Sultan**[2], **Adnan Ahmad Khan**[2,3]*

**1** Akhter Hameed Khan Foundation, Islamabad, Pakistan, **2** Ministry of National Health Services, Regulation and Coordination, Islamabad, Pakistan, **3** Research and Development Solutions, Islamabad, Pakistan

* adnan@resdev.org

**Data Availability Statement:** The data was provided to the Akhter Hameed Khan Foundation team for this study as part of its work with Pakistan's Federal Ministry of National Health

## Abstract

Schools were closed all over Pakistan on November 26, 2020 to reduce community transmission of COVID-19 and reopened between January 18 and February 1, 2021. However, these closures were associated with significant economic and social costs, prompting a review of effectiveness of school closures to reduce the spread of COVID-19 infections in a developing country like Pakistan. A single-group interrupted time series analysis (ITSA) was used to measure the impact of school closures, as well as reopening schools, on daily new COVID-19 cases in 6 major cities across Pakistan: Lahore, Karachi, Islamabad, Quetta, Peshawar, and Muzaffarabad. However, any benefits were contingent on continued closure of schools, as cases bounced back once schools reopened. School closures are associated with a clear and statistically significant reduction in COVID-19 cases by 0.07 to 0.63 cases per 100,000 population, while reopening schools is associated with a statistically significant increase. Lahore is an exception to the effect of school closures, but it too saw an increase in COVID-19 cases after schools reopened in early 2021. We show that closing schools was a viable policy option, especially before vaccines became available. However, its social and economic costs must also be considered.

## Introduction

Since the beginning of the global spread of COVID-19 and before effective vaccines became available, non-pharmaceutical interventions (NPIs), either barriers or means to limit contact between individuals, were the mainstay to control the spread of COVID-19. Perhaps the most widely debated among these NPIs was the closure of schools, which drew criticism for the significant social, learning, economic [1, 2], and physical and mental health costs [3–6] associated with them. Notably, these costs are disproportionately borne by already disadvantaged families [1, 7], thereby exacerbating social and economic inequalities [8].

Prior studies suggest that children infected with COVID-19 are often asymptomatic or have mild symptoms identical to other common respiratory infections [9, 10], and yet they can transmit the infection even when they feel well. Children have also been key spreaders in other

Services, Regulations & Coordination (MoNHSR&C) and the National Command & Operation Centre (NCOC) in Islamabad, which lead Pakistan's response to the COVID-19 pandemic. The AHKRC team has provided analytical support to the above entities, and such created knowledge that has directly informed pandemic policy-making in Pakistan. COVID-19 data is compiled and shared in daily National Situation Reports, or Sitreps, by the National Emergency Operation Centre (NEOC). Each day's Sitrep is compiled as a PDF file. The data used for this study was manually compiled from these PDF files and then used in STATA. The parentage of this data is with the NCOC and the MoNHSR&C. The AHKRC team received this data with the express understanding that it would be kept confidential. However, the data can be obtained independently from the NEOC, through a data request procedure, which is subject to approval from the MoNHSR&C. The data request form as well as the standard operating procedures for a data request are provided as individual documents in the attachments for this submission. The data request itself is to be addressed to: Dr. Shahzad Baig, National Coordinator, National Emergency & Operation Center, D Block, EPI Building, Chak Shahzad, Park Road, Islamabad. Email: eocpakistan@gmail.com Phone: +92-51-8730879.

**Funding:** This work was supported, in whole or in part, by the Bill & Melinda Gates Foundation [grant number: INV-025171]. The funders had no role in study design, data collection and analysis, decision to publish, or preparation of the manuscript.

**Competing interests:** The authors have declared that no competing interests exist.

respiratory infections such as influenza, because of prolonged contact in close proximities with other children at schools [11].

Early evidence on the effect of school closures on epidemic transmission of COVID-19 seemed mixed. Initial, and often modeling-based studies, suggested that closing schools may not help reduce COVID-19 transmission in communities [12–15]. However, more recent, and more empirically based studies have tended to show a role for school closure in reducing cases in the community [16–23]. In low- and middle-income countries such as Pakistan where learning is already inadequate and remote learning solutions are all too often unavailable for most students [24], it is paramount that such a social policy be used only if absolutely supported by evidence of a benefit in limiting COVID-19 transmission and then too, only as a means of last resort. We explore the changes in daily cases on COVID-19 pre and post school closures in Pakistan using a single-group Interrupted Time Series Analysis (ITSA).

This paper is a continuation of our earlier work, which examined the effects of school closures on the daily cases of COVID-19 in Islamabad vs. Peshawar, during the same period as in this study [25]. However, this study attempts to examine the effect of school closures with a different methodology, and also with a larger sample of cities.

## Methods

In this paper we conduct a pre- and post-school closures and reopening analysis of changes in the daily incidence of COVID-19 cases (per 100,000 population) in 6 cities of Pakistan: Lahore, Karachi, Islamabad, Quetta, Peshawar, and Muzaffarabad using a single-group ITSA. These cities are provincial capitals except Islamabad which is the federal capital of Pakistan. We choose these cities for the analysis as these are the largest cities of relevant provinces in Pakistan and much of the COVID-19 caseload was concentrated in these cities during the analysis period. To estimate treatment effects of school closures and reopening, we use a single-group ITSA because it is a quasi-experimental tool that is particularly useful when data cannot be fully randomized, there is no comparison group, and there is a need to consider the effect of only one intervention.

This suits our study as, in Pakistan, all non-school NPIs were enacted in groups–except for the closure of schools. For example, marriage hall restrictions and ban on large scale gatherings were notified at the same time, as were mask-wearing, broader "smart" lockdowns (lockdowns in parts of cities), and reduced market timings. Mask-wearing and social distancing were constant across time with similar compliance across the country. Two major non-school NPIs that were implemented in these cities during our period of study were marriage hall restrictions and smart lockdowns (see S1 Table). These NPIs targeted only a small proportion of the population and hence their impact was assumed to be limited. School closures, on the other hand, were universally enforced and applied to all schools–public or private, day or boarding–and to students of all grades across Pakistan [26].

Data for this analysis was sourced from the daily National Situation Reports (Sitreps) published by the National Emergency Operations Centre (NEOC) in Islamabad, Pakistan. It is the only Ministry of Health's official COVID-19 data that was being analyzed and used for Pakistan's pandemic response, nationally and internationally. This data is anonymized and aggregated by city, with no disaggregation by age, gender, ethnicity, or any other potentially identifying characteristic. We use this data for an inferential analysis of the change in daily COVID-19 incidence in the overall populations of the 6 aforementioned cities, regardless of demographic characteristics, due to the change in one particular NPI. It is because this NPI is the only policy intervention that could be isolated in our chosen time period of observation. Populations of cities were derived from Population Census 2017 of Pakistan, which is publicly available [27].

**Table 1. Key intervention dates of school closures and reopening.**

|  | School Closures | School Reopening |
|---|---|---|
| **Original** | November 26, 2020 | February 1, 2021 |
| **With 10-day delay** | December 6, 2020 | February 11, 2021 |
| **With 20-day delay** | December 16, 2020 | February 21, 2021 |

Note: 2 dates were removed from analysis: 4th November 2020 for Quetta and 7th December 2020 for Muzaffarabad because no data for these dates was available.

We estimated 2 sets of ordinary least square (OLS) regressions for each city using a 10- or 20-day delay since COVID-19 incidence changes from school-related NPIs take effect 10 [28] or more days [17, 29, 30] after closures or reopening. Daily new COVID-19 cases (per 100,000 population) were taken for equally spaced time frames with 10- and 20-days delay after the actual school closures and reopening dates. In order to analyze the effect of school closures and reopening, we took a total of 60 days for pre- and post-intervention periods. Table 1 shows the dates of school closures and reopening. For 10-days delay models of school closures and reopening, we took the following dates from November 06, 2020 to January 04, 2021 and January 12, 2021 to March 12, 2021, respectively. For 20-days delay models of school closures and reopening, we took the time periods from November 16, 2020 to January 14, 2021 and January 22, 2021 to March 22, 2021 respectively. We chose a 60-days period because a sample size of at least 50 time points is recommended by Box and Jenkens [31], as larger sample size increases power in the case of segmented time series [32]. We did not take a much longer time period since schools had started to reopen after one-month post-closures period in our 20-days delay model [33].

## Model specification

Our model specification is adapted from Linden and detailed below [34]:

$$Y_{ti} = \beta_0 + \beta_1 T_{ti} + \beta_2 X_{ti} + \beta_3 X_{ti} T_{ti} + \epsilon_{ti}$$

Where

1. $Y_{ti}$, our outcome variable, is the daily number of new COVID-19 cases (per 100,000 population) in city $i$;

2. $\beta_0$, the constant term, is the starting level of the daily new COVID-19 cases (per 100,000 population) in city $i$;

3. $T_{ti}$ is the time period since the beginning of this study, and the coefficient $\beta_1$ shows the slope of daily new COVID-19 cases (per 100,000 population) until the start of the intervention for city $i$;

4. $X_{ti}$ is a dummy variable indicating the intervention period (post intervention = 1, and 0 otherwise) for city $i$;

5. $\beta_2$ explains the change in daily COVID-19 cases (per 100,000 population) that occurs in the time period immediately followed by the school closure/reopening (our interventions) in city $i$;

6. $X_{ti}T_{ti}$ is the interaction term between the intervention period and the time since the start of the study; and,

7. **β₃** represents the difference between the pre- and post-intervention slopes for daily new COVID-19 cases (per 100,000 population) in city $i$.

We run the model specified above for each city $i$ separately. Since we want to estimate city-level effects as compared to a national or combined effect, we do not opt for a panel data model. To get a singular, direct estimate of the effect of closing/reopening schools, we used the *lincom* estimate which is the sum of $\beta_1$ and $\beta_3$ [35].

$$Treated = \beta_1 T_{ti} + \beta_3 X_{ti} T_{ti}$$

This generates a separate variable that sums the values of β₁ and β₃.

To adjust for autocorrelation and possible heteroskedasticity, we used Newey-West standard errors in our regression models [36]. Cumby-Huizinga test for autocorrelation was performed on each regression model to identify correct lag structure. Linktest was applied to check if the models were correctly specified. Stata 16 software package was used for the analysis.

## Results

Our descriptive results showing means and standard deviations of daily COVID-19 cases (per 100,000 population) are presented in Table 2. These were calculated for school closures and reopening periods of each city, separately for pre- and post-intervention periods. For 10-day delay, Islamabad showed highest cases (per 100,000 population) (18.80, SD: 4.411), while Quetta showed the lowest (0.722, SD: 0.294) in the pre-intervention period of school closures; same trend followed in school reopening pre-intervention period. Islamabad and Quetta also had the highest and the lowest cases (per 100,000 population) in pre-intervention periods of 20-day delay school closures and reopening.

After adding a 10-day delay after the actual date of school closures (Table 3), the rate of change in daily COVID-19 cases declined following closure of schools in Karachi, Islamabad, Quetta, and Peshawar; the reductions per 100,000 population were by -0.16 cases (95% CI: -0.23, -0.13) in Karachi, -0.41 cases (95% CI: -0.53, -0.30) in Islamabad, -0.01 cases (95% CI: -0.01, -0.00) in Quetta, and -0.06 cases (95% CI: -0.08, -0.03) in Peshawar. In Lahore, daily

**Table 2. Summary statistics of daily COVID-19 cases (per 100,000 population) in each city.**

| | City | 10-days Delay | | 20-days Delay | |
|---|---|---|---|---|---|
| | | Pre-intervention | Post-intervention | Pre-intervention | Post-intervention |
| **School Closures** | Lahore | 1.23 (0.34) | 1.62 (0.48) | 1.32 (0.34) | 1.9 (0.39) |
| | Karachi | 5.53 (2.07) | 6.66 (2.13) | 7.26 (1.9) | 5.77 (1.66) |
| | Islamabad | 18.8 (4.41) | 10.4 (4.42) | 18.4 (4.7) | 7.62 (2.12) |
| | Quetta | 0.72 (0.29) | 0.26 (0.11) | 0.6 (0.3) | 0.22 (0.1) |
| | Peshawar | 1.59 (0.76) | 2.18 (0.94) | 2.11 (0.86) | 1.93 (0.81) |
| | Muzaffarabad | 4.58 (2.54) | 0.82 (0.79) | 3.11 (2.41) | 0.52 (0.61) |
| **School Reopening** | Lahore | 1.64 (0.43) | 2.130 (0.88) | 1.48 (0.34) | 3.35 (1.55) |
| | Karachi | 3.94 (2.03) | 1.045 (0.37) | 2.38 (1.25) | 0.86 (0.28) |
| | Islamabad | 4.9 (1.26) | 7.945 (3.79) | 4.74 (0.98) | 14.6 (9.34) |
| | Quetta | 0.16 (0.08) | 0.145 (0.13) | 0.12 (0.07) | 0.22 (0.13) |
| | Peshawar | 1.55 (0.55) | 1.248 (0.49) | 1.28 (0.41) | 1.9 (1.14) |
| | Muzaffarabad | 0.19 (0.25) | 0.55 (0.63) | 0.19 (0.23) | 1.22 (1.34) |

Note: Means of daily COVID-19 cases over each period are reported. Standard deviations are in parentheses.

**Table 3. Rates of change in daily COVID-19 cases (per 100,000 population) through Interrupted Time Series Analysis.**

| City | Delay after original date of intervention | Rates of Change in Daily COVID-19 cases (per 100,000 population) (95% CI), p-value | | | | | |
|---|---|---|---|---|---|---|---|
| | | School Closure | | | School Reopening | | |
| | | (1) Pre-Closure trend: $\beta_1$ | (2) Post-Closure trend: $\beta_1$-$\beta_3$ | (3) Closure Difference: $\beta_3$ | (1) Pre-Reopening trend: $\beta_1$ | (2) Post-Reopening trend: $\beta_1$-$\beta_3$ | (3) Reopening Difference: $\beta_3$ |
| **Lahore** | 10-day Delay | 0.02* (0.01, 0.04) p<0.01 | 0.03* (0.01, 0.05) p<0.01 | 0.00 (-0.02, 0.03) | -0.03* (-0.05, -0.02) p<0.01 | 0.09* (0.06, 0.11) p<0.01 | 0.12* (0.09, 0.15) p<0.01 |
| | 20-day Delay | -0.00 (-0.00, 0.00) | 0.01* (0.00, 0.02) p<0.05 | 0.01* (0.00, 0.02) p<0.05 | -0.02* (-0.04, 0.00) p<0.01 | 0.16* (0.12, 0.20) p<0.01 | 0.18* (0.14, 0.23) p<0.01 |
| **Karachi** | 10-day Delay | 0.22* (0.18, 0.26) p<0.01 | -0.16* (-0.23, -0.13) p<0.01 | -0.39* (-0.46, -0.33) p<0.01 | -0.20* (-0.24, -0.16) p<0.01 | -0.03* (-0.04, -0.02) p<0.01 | 0.17* (0.12, 0.21) p<0.01 |
| | 20-day Delay | 0.16* (0.07, 0.25) p<0.01 | 0.00 (-0.10, 0.10) | -0.16* (-0.26, -0.05) p<0.01 | -0.11* (-0.14, -0.08) p<0.01 | -0.01 (-0.02, 0.00) p<0.10 | 0.10* (0.07, 0.13) p<0.01 |
| **Islamabad** | 10-day Delay | 0.21* (0.07, 0.36) p<0.01 | -0.41* (-0.53, -0.30) p<0.01 | -0.63* (-0.81, -0.44) p<0.01 | -0.08* (-0.12, -0.04) p<0.01 | 0.34* (0.18, 0.50) p<0.01 | 0.42* (0.24, 0.60) p<0.01 |
| | 20-day Delay | -0.22 (-0.48, 0.05) | -0.12* (-0.21, -0.03) p<0.05 | 0.10 (-0.18, 0.38) | 0.05* (0.00, 0.09) p<0.05 | 0.98* (0.69, 1.26) p<0.01 | 0.93* (0.63, 1.23) p<0.01 |
| **Quetta** | 10-day Delay | 0.01 (-0.01, 0.02) | -0.01* (-0.01, -0.00) p<0.01 | -0.01 (-0.03, 0.01) | -0.01* (-0.01, -0.00) p<0.01 | 0.01* (0.00, 0.01) p<0.01 | 0.01* (0.01, 0.02) p<0.01 |
| | 20-day Delay | -0.02* (-0.03, -0.01) p<0.01 | -0.00 (-0.01, 0.00) | 0.02* (0.01, 0.03) p<0.01 | -0.00* (-0.01, -0.00) p<0.01 | 0.01* (0.00, 0.01) p<0.01 | 0.01* (0.00, 0.02) p<0.01 |
| **Peshawar** | 10-day Delay | 0.05* (0.03, 0.08) p<0.01 | -0.06* (-0.08, -0.03) p<0.01 | -0.11* (-0.14, -0.08) p<0.01 | -0.04* (-0.06, -0.04) p<0.01 | 0.02* (0.01, 0.04) p<0.01 | 0.06* (0.05, 0.08) p<0.01 |
| | 20-day Delay | 0.04* (0.01, 0.07) p<0.01 | -0.03 (-0.06, 0.01) | -0.07* (-0.11, -0.02) p<0.01 | -0.03* (-0.04, -0.02) p<0.01 | 0.10* (0.06, 0.14) p<0.01 | 0.13* (0.08, 0.18) p<0.01 |
| **Muzaffarabad** | 10-day Delay | -0.11* (-0.16, -0.06) p<0.01 | -0.05* (-0.06, -0.03) p<0.01 | 0.06* (0.01, 0.12) p<0.05 | 0.00 (-0.01, 0.01) p<0.01 | 0.04* (0.02, 0.06) p<0.01 | 0.04* (0.01, 0.06) p<0.01 |
| | 20-day Delay | -0.17* (-0.26, -0.07) p<0.01 | -0.03 (-0.06, 0.00) | 0.14* (0.04, 0.24) p<0.01 | 0.01* (0.00, 0.01) p<0.01 | 0.09* (0.04, 0.16) p<0.01 | 0.09* (0.03, 0.15) p<0.01 |

*Significant at 95% CI. Newey-West standard errors are used.

COVID-19 cases continued to rise at a rate of 0.03 cases (95% CI: 0.01, 0.05) per 100,000 population after the closure of schools. For Muzaffarabad, the rate of change of COVID-19 cases was declining both before and after the school closure, at -0.11 cases (95% CI: -0.16, -0.06) and -0.05 cases (95% CI: -0.06, -0.03) per 100,000 population, respectively.

The opposite trend was seen following schools reopening in early 2021. Before schools reopened, accounting for a 10-day delay from the actual date of reopening, the rate of change of daily COVID-19 cases per 100,000 population was declining in every city by: -0.03 cases (95% CI: -0.05, -0.02) in Lahore, -0.20 cases (95% CI: -0.24, -0.16) in Karachi, -0.08 cases (95% CI: 0.12, -0.04) in Islamabad, -0.01 cases (95% CI: -0.01, -0.00) in Quetta, and by -0.04 cases

(95% CI: -0.06, -0.04) in Peshawar. Muzaffarabad's pre-reopening trend is not statistically significant at 95% confidence level.

After schools reopened, the rate of change of daily COVID-19 cases became positive in every city–except in Karachi, where the rate remained negative at -0.03 cases (95% CI: -0.04, -0.02) per 100,000 population. For the remaining cities, daily new COVID-19 cases began to increase at a rate of 0.09 cases (95% CI: 0.06, 0.11) per 100,000 population in Lahore, 0.34 cases (95% CI: 0.18, 0.50) in Islamabad, 0.01 cases (95% CI: 0.00, 0.01) in Quetta, 0.02 cases (95% CI: 0.01, 0.04) in Peshawar, and 0.09 cases (95% CI: 0.04, 0.16) in Muzaffarabad.

These effects were similar but more modest when allowing for a 20-day delay after the actual date of school closures. The post-closure trend remained statistically significant in Lahore, where the rate of change in cases continued to rise at 0.01 cases (95% CI: 0.00, 0.02) per 100,000 population, while Islamabad's post-closure trend showed a decline at a rate of -0.12 (95% CI: -0.21, -0.03). Post-reopening, in Islamabad, the rate of daily new COVID-19 cases changed from 0.05 cases (95% CI: 0.00, 0.09) per 100,000 to 0.98 cases (95% CI: 0.69, 1.26) per 100,000 population while in Muzaffarabad, rate of change went from 0.01 cases (95% CI: 0.00, 0.01) before schools were reopened to 0.09 cases (95% CI: 0.04, 0.16) once schools opened. These results are presented graphically in Figs 1–4 below.

## Discussion

We show that school closures are associated with fewer daily new COVID-19 cases compared to pre-closures by 5 to 62 actual daily cases in individual cities. Correspondingly, reopening schools appear to increase them by 1 to 35 daily cases. The association is the strongest in Karachi, Lahore, Peshawar, and Islamabad [37], which are larger, denser cities, and had the most

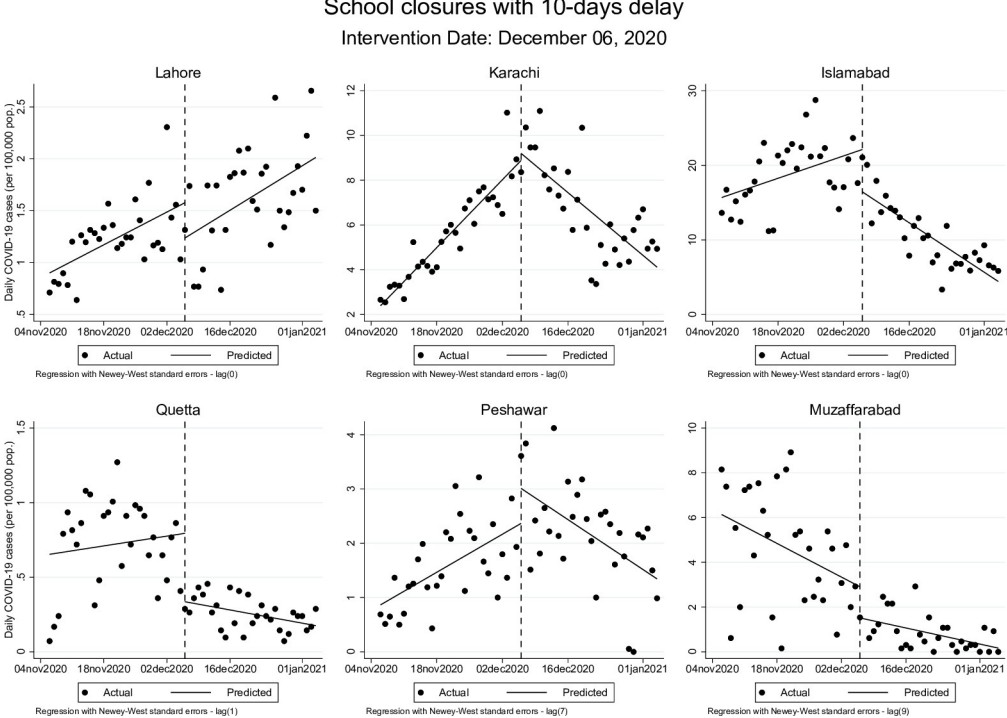

**Fig 1. Changes in daily COVID-19 cases (per 100,000 population) during school closures period accounting for 10-days delay.**

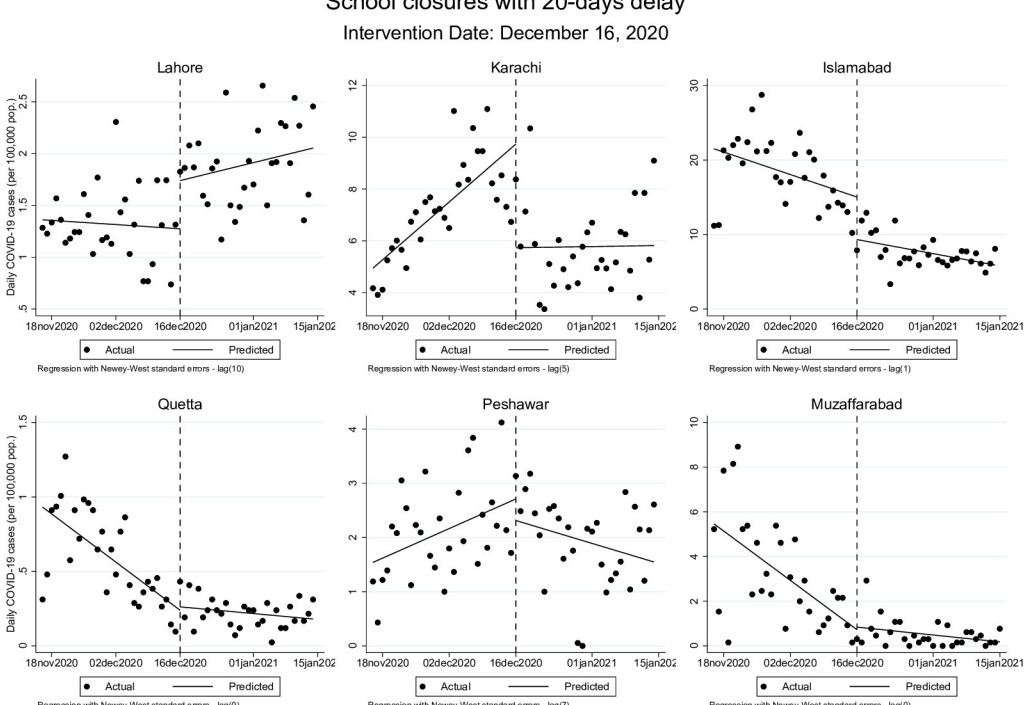

**Fig 2. Changes in daily COVID-19 cases (per 100,000 population) during school closures period accounting for 20-days delay.**

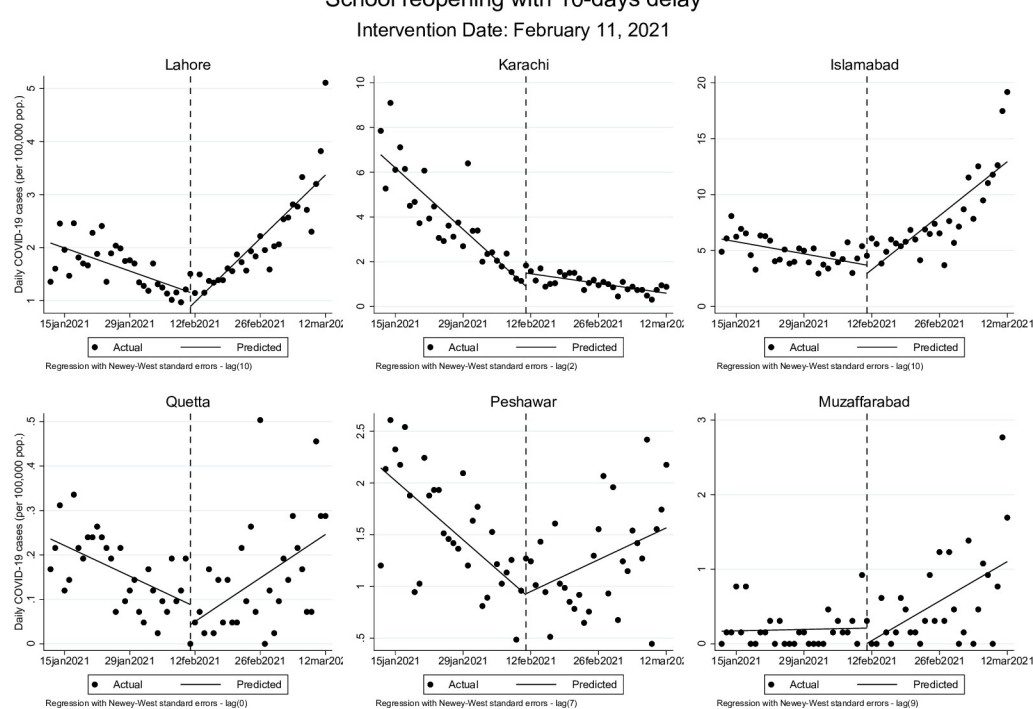

**Fig 3. Changes in daily COVID-19 cases (per 100,000 population) during school reopening period accounting for 10-days delay.**

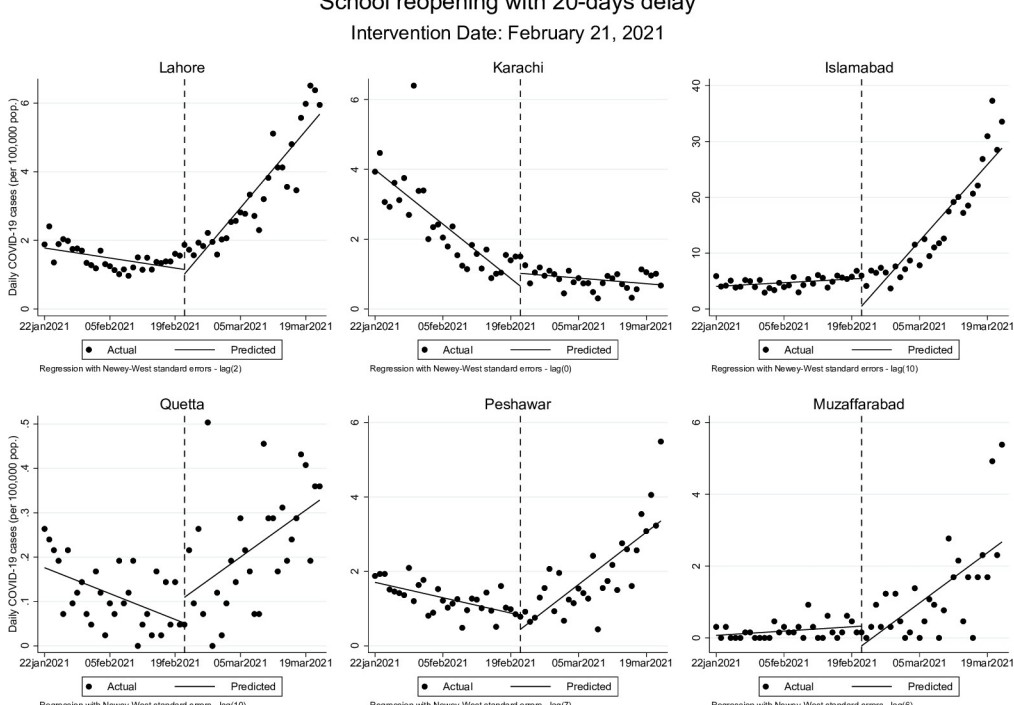

**Fig 4. Changes in daily COVID-19 cases (per 100,000 population) during school reopening period accounting for 20-days delay.**

cases; while the association was modest for sparse and smaller cities of Quetta and Muzaffara-bad, that also had fewer overall cases.

Pakistan saw a much lower reduction in COVID-19 cases with school closures than was seen in many other countries. For example, our reduction of 5 to 62 daily cases is considerably smaller than the reduction of 424 cases per 100,000 population seen in the USA [17]. However, this is consistent with the fact that cities with fewer cases had the least reductions in cases when schools were closed. Pakistan has also seen much fewer cases, hospitalizations, and deaths from COVID-19 than in Europe or North America, perhaps relating to its sparser social networks leading to fewer contacts among individuals within the community [38], or perhaps higher levels of nonspecific immunity from prior infections with disparate viruses [39]. However, both of these possible explanations are speculative at the moment.

Reductions in cases with school closure is better understood in the context of the education set up in Pakistan. Most school-going children in cities go to low-cost-private schools where they sit in small classrooms with little space for social distancing [40]. Additionally, most children commute to and from schools in small vehicles–up to 15–17 children in the back of a minivan. Children then come into contact with adults at school (teachers and custodian staff), then at home (parents and elder family members), and in so doing become a conduit for COVID-19 spread outside of schools. As 31% of total population of Pakistan falls in the school going age [41], schools in Pakistan then essentially function as "super spreader" locations for COVID-19.

Our findings are consistent with the global evidence, as well as the results of our own previous work [25], that school closures are associated with reduction in COVID-19 transmission in communities [17, 30]. In the US, school closures were associated with reduced COVID-19 caseloads [15], deaths [19, 42, 43], and hospitalizations by as much as half [19]. Similarly, the

timely closure of schools and high education institutes were found to lower COVID-19 transmission rates in the European Union and other developed countries [44–47]. Earlier in the epidemic, a number of modeling studies had predicted more modest effects of such closures [19, 21, 24, 44–46]. However, more recent studies using empirical community transmission data have generally shown a more robust association between school closures and reductions in community cases of COVID-19 [21].

### Limitations

There are limitations of this analysis. The daily COVID-19 data are aggregated nationally, regionally, and by certain major cities, with no disaggregation by age or gender. Additionally, we acknowledge that a pre- and post-intervention analysis itself has limitations. For example, it may not effectively discern the effects of an intervention from that of a long-term trend on an outcome variable. This is referred to as the "maturation" threat to the internal validity of a pre- and post-intervention analysis. However, this has negligible impact on our analysis, as we consider a total of 60 days for each ITSA regression, in each of the 6 cities, when examining the effects of the intervention after accounting for sufficient delays–at 10 and 20 days–to be sure of the effects of the interventions.

It is difficult to explain why Lahore did not show any reduction in cases after school closures. It is possible that the epidemic affected cities at different points in time and that it was at a relatively lower level in Lahore during the study period. We also acknowledge that there could be potential cross-contamination of COVID-19 cases between Islamabad and Peshawar, which are separated by a 2-hours commute by road, and between Islamabad and Lahore, which are 4-hours apart by road. However, there are no data on the magnitude of any potential contamination due to bilateral intra-city travel. Nevertheless, were there significant contamination between the cities, one would have expected to see convergence in COVID-19 caseloads between them, and there is no evidence that this occurred. Finally, measurement of the serious social, economic, and educational attainment costs from school closures was beyond the scope of our study.

### Conclusions

School closures may be associated with lower transmission of COVID-19 in communities and such closures are an important policy tool to stop the spread of COVID-19. However, their social and economic costs are high, perhaps more so in a developing country. The balance of these costs and benefits must inform this effective NPI specially when other measures, including vaccines, are being planned.

### Supporting information

**S1 Table. Non-school closures non-pharmaceutical interventions (NPIs) during study period (November 6, 2020 to March 22, 2021).**
(PDF)

**S1 Text. Methodology steps.**
(DOCX)

**S1 Data. Dataset for 10-days delay.**
(DTA)

**S2 Data. Dataset for 20-days delay.**
(DTA)

## Acknowledgments

We thank Testing, Tracing and Quarantining (TTQ) team at the National Command and Operation Centre (NCOC) and the National Emergency Operations Centre (NEOC) for facilitating our work.

## Author Contributions

**Conceptualization:** Abdul Mueed, Mujahid Abdullah, Adnan Ahmad Khan.

**Data curation:** Taimoor Ahmad, Mujahid Abdullah.

**Formal analysis:** Abdul Mueed, Taimoor Ahmad, Mujahid Abdullah.

**Funding acquisition:** Adnan Ahmad Khan.

**Investigation:** Abdul Mueed, Taimoor Ahmad, Mujahid Abdullah, Faisal Sultan, Adnan Ahmad Khan.

**Methodology:** Abdul Mueed, Taimoor Ahmad, Mujahid Abdullah.

**Project administration:** Abdul Mueed, Adnan Ahmad Khan.

**Resources:** Taimoor Ahmad, Faisal Sultan, Adnan Ahmad Khan.

**Software:** Taimoor Ahmad, Mujahid Abdullah.

**Supervision:** Faisal Sultan, Adnan Ahmad Khan.

**Validation:** Faisal Sultan, Adnan Ahmad Khan.

**Visualization:** Taimoor Ahmad, Mujahid Abdullah.

**Writing – original draft:** Abdul Mueed, Taimoor Ahmad.

**Writing – review & editing:** Abdul Mueed, Taimoor Ahmad, Mujahid Abdullah, Faisal Sultan, Adnan Ahmad Khan.

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
