## [Decision Letter · Decision Letter 0]

6 Jul 2022

PGPH-D-22-00853

Impact of school closures and reopening on COVID-19 caseload in 6 cities of Pakistan: An Interrupted Time Series Analysis

Dear Dr. Khan,

Thank you for submitting your manuscript to PLOS Global Public Health. After careful consideration, we feel that it has merit but does not fully meet PLOS Global Public Health’s publication criteria as it currently stands. Therefore, we invite you to submit a revised version of the manuscript that addresses the points raised during the review process.

We look forward to receiving your revised manuscript.

Kind regards,

Xerxes Tesoro Seposo, MPH, PhD

Academic Editor

Journal Requirements:

Additional Editor Comments (if provided):

Dear Dr. Adnan Khan,

Please address in full the queries raised by the respective reviewers, in particular a more elaborate explanation and justification of the data as well as the parameterization of the method used. Furthermore, please adhere to the specifications set by the journal in order to facilitate a more coherent processing of the manuscript.

Kind Regards,

Xerxes Seposo

Academic Editor

PloS Global Public Health

Reviewers' comments:

Reviewer's Responses to Questions

**Comments to the Author**

1. Does this manuscript meet PLOS Global Public Health’s publication criteria? Is the manuscript technically sound, and do the data support the conclusions? The manuscript must describe methodologically and ethically rigorous research with conclusions that are appropriately drawn based on the data presented.

Reviewer #1: Yes

Reviewer #2: Yes

2. Has the statistical analysis been performed appropriately and rigorously?

Reviewer #1: Yes

Reviewer #2: Yes

3. Have the authors made all data underlying the findings in their manuscript fully available (please refer to the Data Availability Statement at the start of the manuscript PDF file)?

Reviewer #1: Yes

Reviewer #2: Yes

4. Is the manuscript presented in an intelligible fashion and written in standard English?

Reviewer #1: Yes

Reviewer #2: Yes

5. Review Comments to the Author

Reviewer #1: 1. Please describe your reasons for choosing only provincial capitals for data analyses?

2. why only 60 days of window pre and post for data analyses?

3. please provide which schools were included and why??

Reviewer #2: Review: Impact of school closures and reopening on COVID-19 caseload in 6 cities of Pakistan:

An Interrupted Time Series Analysis

Decision: Revision, consider for publication after changes!

Overview: The paper seeks to report that school closure contributed to reduction in covid cases in six cities of Pakistan. The subject of course is an important one and perhaps somewhat difficult to estimate given that in these periods most countries had adopted a number of measures to combat Covid 19. The authors use single group interrupted time series model to detect changes that may have been brought about by school closure. The paper can be improved to be accepted into an international journal. Authors should copy edit.

Specific Comments: Much of your arguments for writing this paper hinges on the second paragraph in the methods section. I don’t understand the second sentence in that paragraph. Is it saying that all cities also had several restrictions at the same time as was the case with school closures? It seems to me that everyone applied all the programs at the same time. Are you saying that school closures fluctuated during this time in all six cities where as other measures did not? I would like to see a greater defense of the line that NPI is the only policy intervention that could be isolated in your chose time period of observation to be consistent across six counties. So, from Table 1, should we understand that all other programs do not exist during these times? Auger et al. paper incorporates a lag period to allow for potential policy-associated change to occur. Your reasoning for delays is different, 1st line 4th para in Methods section. You have only six cities, can you state what other policies might be in place during these times? You could introduce a table stating what was in place during the times listed in Table 1 for each city.

Can you state how reliable is your data. Please justify the data set and also state its limitations.

Your equation specification reads like a panel. You may think of clarifying it. You run six estimations for each city, not a panel. It would be nice to have graphical representation of all six cities and also the country using the raw data. You might want to estimate your results in terms of percentages in the discussion section. I don’t follow how you obtained the last line in the 1st para of the Discussion section. I am puzzled by the reference to 34, the Doug North et al. book. Do you mean social networks contacting each other? It is likely that personal communication between relatives would be higher in Pakistan and in the cities they would be dispersed? Comparison should be with other South Asia and Vietnam as Europe and North America have much greater resources.

I am not sure school closure impact is conclusive even in the works by Auger et al. Identification strategy amid so many other restriction is suggestive at best, especially since alternate policies were likely to be many during these times.

6. PLOS authors have the option to publish the peer review history of their article (what does this mean?). If published, this will include your full peer review and any attached files.

**Do you want your identity to be public for this peer review?** For information about this choice, including consent withdrawal, please see our Privacy Policy.

Reviewer #1: **Yes: **Faisal Abbas

Reviewer #2: **Yes: **Arnab K Acharya

---

## [Editor Report · Decision Letter 1]

26 Aug 2022

Impact of school closures and reopening on COVID-19 caseload in 6 cities of Pakistan: An Interrupted Time Series Analysis

PGPH-D-22-00853R1

Dear Dr Khan,

We are pleased to inform you that your manuscript 'Impact of school closures and reopening on COVID-19 caseload in 6 cities of Pakistan: An Interrupted Time Series Analysis' has been provisionally accepted for publication in PLOS Global Public Health.

Best regards,

Xerxes Tesoro Seposo, MPH, PhD

Academic Editor
